# Improved Genetic Characterization of Hypercholesterolemia in Latvian Patients with Familial Hypercholesterolemia: A Combined Monogenic and Polygenic Approach Using Whole-Genome Sequencing

**DOI:** 10.3390/ijms252413466

**Published:** 2024-12-16

**Authors:** Ivanna Atava, Monta Briviba, Georgijs Nesterovics, Vita Saripo, Dainus Gilis, Ruta Meiere, Elizabete Terauda, Gunda Skudrina, Janis Klovins, Gustavs Latkovskis

**Affiliations:** 1Latvian Biomedical Research and Study Centre, LV-1067 Riga, Latvia; ivanna.atava@biomed.lu.lv (I.A.); monta.briviba@biomed.lu.lv (M.B.); klovins@biomed.lu.lv (J.K.); 2Institute of Cardiology and Regenerative Medicine, University of Latvia, LV-1004 Riga, Latvia; gnesterovics@gmail.com (G.N.); vita.saripo@inbox.lv (V.S.); dainus.gilis@gmail.com (D.G.); ruuta.meiere@gmail.com (R.M.); elizabete.terauda@gmail.com (E.T.); 3Faculty of Medicine and Life Sciences, University of Latvia, LV-1004 Riga, Latvia; 4Latvian Center of Cardiology, Pauls Stradins Clinical University Hospital, LV-1002 Riga, Latvia; gunda.skudrina@inbox.lv

**Keywords:** familial hypercholesterolemia, severe hypercholesterolemia, low-density lipoprotein cholesterol, lipoprotein (a), whole-genome sequencing, polygenic risk score, monogenic hypercholesterolemia

## Abstract

Despite the implementation of next-generation sequencing-based genetic testing on patients with clinical familial hypercholesterolemia (FH), most cases lack complete genetic characterization. We aim to investigate the utility of the polygenic risk score (PRS) in specifying the genetic background of patients from the Latvian Registry of FH (LRFH). We analyzed the whole-genome sequencing (WGS) data of the clinically diagnosed FH patients (n = 339) and controls selected from the Latvian reference population (n = 515). Variant pathogenicity in FH patients was classified according to the ACMG/AMP guidelines. The low-density lipoprotein cholesterol (LDL-C) and lipoprotein (a) (LPA) PRS were calculated based on the WGS data. We identified unique causative variants in 80 (23.6%) of the tested individuals (39 variants in FH genes and 4 variants in phenocopy genes, with 6 variants being novel). The LDL-C PRS was highly discriminative compared to the LPA PRS. Nevertheless, both PRS were able to explain the genetic cause of hypercholesterolemia in 26.3% of the remaining non-monogenic patients. The combined genetic analysis of monogenic and polygenic hypercholesterolemia resulted in 43.7% genetically explained hypercholesterolemia cases. Even though the application of PRS alone does not exclude monogenic testing in clinical FH patients, it is a valuable tool for diagnosis specification.

## 1. Introduction

Familial hypercholesterolemia (FH) is one of the most common autosomal dominant genetic disorders (estimated prevalence of 1:200–313 and 1:160,000–300,000 for heterozygous (HeFH) and homozygous (HoFH) forms, respectively [1,2,3]. It is characterized by an increased low-density lipoprotein cholesterol (LDL-C) level, which leads to premature atherosclerosis and a high risk of cardiovascular diseases (CVD) [1]. The loss of function (LoF) variants in *LDLR* (most prevalent) and *APOB*, the gain of function variants in *PCSK9* genes, and, rarely, the homozygous LoF variations in the *LDLRAP1* gene result in the FH phenotype, presenting in two forms [4]. The HoFH is characterized by biallelic (homozygous or compound heterozygous) variants, resulting in a much more severe phenotype than the HeFH [5].

Familial hypercholesterolemia in Latvia is diagnosed primarily using the Dutch Lipid Clinic Network (DLCN) criteria through opportunistic screening of adults and cascade screening of first-degree relatives of index cases [6]. Universal FH screening is not implemented. The Latvian Registry of FH (LRFH), established in 2015, aims to improve FH diagnosis and management. It focuses on identifying individuals, initiating early treatment, and implementing cascade screening. While genetic testing is not routinely available, genetic studies are conducted based on research grants, with many FH patients participating in the Genome Database of Latvian Population (LGDB) [6].

Earlier, based on the cohort of the LRFH, we made the first steps in describing the genetic background of Latvian FH patients, and we were able to identify and describe causal variations in 20.9% of cases [7]. However, the explanation for clinical FH phenotype in a major part of our cohort remains unknown.

Polygenic hypercholesterolemia (PH) is expected to make up to approximately 30% of patients with clinically assigned FH [8]. In those patients, the combination of small-effect alleles could result in LDL-C levels similar to the monogenic form, complicating the clinical distinction. One approach in determining PH is to apply the LDL-C polygenic risk score (PRS) to a cohort in which preliminary genetic testing for a monogenous form was performed. This is crucial for the consequent diagnostic differentiation of hypercholesterolemia [9], as high PRS was previously reported both in polygenic and monogenic patients (with usually greater LDL-C and increased CVD risk in the latter group) [10]. The FH phenotype has also been attributed to elevated levels of lipoprotein (a) (Lp(a)) and PRS (LPA PRS) [11].

Therefore, given the utility of PRS to reveal additional (to those with monogenic origin) patients with the genetic cause of hypercholesterolemia, we aimed to apply the LDL-C and LPA PRS approaches to the obtained whole-genome sequencing (WGS) data to inspect the burden of PH and Lp(a) in the Latvian FH cohort. Before that, we investigated and characterized the rare genetic variants that result in monogenic FH in all patients included in the study.

## 2. Results

### 2.1. Cohort Characteristics

Among 339 cohort patients, 132 (38.9%) were males; the mean age (SD) was 51.5 ± 12.1 (range 23–79 years); 176 (51.9%) had a history of coronary artery disease (CAD), 123 (36.3%) of premature CAD, 33 (9.7%) of myocardial infarction, and 56 (16.5%) of coronary revascularization (Table 1). The highest documented LDL-C and total cholesterol levels were 7.1 ± 1.5 mmol/L and 9.4 ± 2.3 mmol/L, respectively. The clinical diagnoses based on the DLCN criteria were the following: 87 (25.7%): definite FH, 153 (45.1%): probable FH, and 99 (29.2%): possible FH. The patients had the following risk factors and physical signs: arterial hypertension (n = 148, 43.7%); diabetes (n = 14, 4.1%); obesity, defined as body mass index (BMI) ≥ 30 kg/m^2^ (n = 64, 18.9%); mean BMI 27.3 ± 4.0 kg/m^2^; current or previous smoking (n = 117, 34.5%); tendon xanthomas (n = 55, 16.2%); arcus cornealis before age 45 (n = 10, 2.9%); and xanthelasmas (n = 15, 4.4%). The characteristics of the clinical FH patients based on genetic findings are shown in Table 1, with specifications for patients of non-FH origin shown in Appendix A.

Among the 515 control group individuals, 157 (30.5%) were males; the mean age (SD) was 40.0 ± 11.7 (range 18–84 years). The mean BMI (SD) was 25.6 ± 4.8 kg/m^2^, and 88 (17.1%) individuals were obese.

### 2.2. Causative Genetic Variants

Out of 339 index cases with successful WGS, 80 patients had 43 unique pathogenic (P) and likely pathogenic (LP) variants. These included 27 missense, 1 synonymous, 3 splice sites, 6 nonsense single nucleotide polymorphisms (SNPs), and 6 structural variants (SVs, 5 frameshifts) (Table 2). The majority of the patients (n = 73) had heterozygous variants, while three were homozygous (of these, two patients had variants in the FH phenocopy gene *LIPA*), and one patient was compound heterozygous. Of the 43 variants, 39 were found in “standard” FH genes, and the remaining ones were found in genes associated with other dyslipidemia-related conditions.

Out of 43 P and LP variants, 6 were novel: p.Arg3527Gly in *APOB*; p.Ala431Val, c.1061-2A>C, c.1846-2A>G, and p.Lys411SerfsTer4 in *LDLR*; and p.Thr151LeufsTer83 in *ABCG5*. Only the c.1846-2A>G variant was present in the Genome Aggregation Database (gnomAD) database [12] (at extremely low frequency); other variations were absent from this database.

The *APOB* missense variant p.Arg3527Gly is located in a codon that is well-known for two other pathogenic variants—p.Arg3527Gln (also found in our study) and p.Arg3527Trp. It was discussed before that the replacement of Arg3527 results in conformational changes and significantly reduces the apoB100 affinity for the low-density lipoprotein (LDL) receptor [13], which indicates that this codon is critical for protein function. The patient with this variant was diagnosed with probable FH (DLCN score = 6).

The *LDLR* missense variant, p.Ala431Val, was found in two patients with probable FH in our cohort and previously reported in the Russian population [14], although, to our knowledge, it was not reported in any other database. There is another P and two LP variations (p.Ala431Thr, p.Ala431Pro, and p.Ala431Gly, respectively) in the same codon reported. In silico algorithms predict this variant to be deleterious (REVEL = 0.89).

Both the *LDLR* splice site variants (c.1061-2A>C and c.1846-2A>G) disrupt the acceptor splice site, although there are no studies on these variants’ predicted effect on the reading frame, and to stay conservative, we applied the PVS1_Strong criteria.

Two novel frameshift insertions, p.Lys411SerfsTer4 in *LDLR* and p.Thr151LeufsTer83 in the *ABCG5* gene, are predicted to undergo nonsense-mediated mRNA decay. Both variations were found in patients with a definite FH diagnosis (DLCN score > 8).

In our study, one case with compound heterozygous FH was found. The index patient with definite FH (diagnosed as clinical homozygous FH) was carrying a missense variant p.Ala540Thr and a frameshift duplication p.Val806GlyfsTer11. The highest documented LDL-C was 14.39 mmol/L and the triglycerides level (at the same time) was 1.82 mmol/L, and the patient was resistant to all three anti-PCSK9 therapies (evolocumab, alirocumab, and inclisiran) when added to rosuvastatin 40 mg and ezetimibe 10 mg. The patient also had a family history of premature CAD. Two patients’ offspring (son and daughter) were included in this study and underwent WGS, which revealed that both are the carriers of the p.Ala540Thr variant in a heterozygous state (Table 2). They had a much less severe phenotype (with the highest documented LDL-C levels of 6.70 and 5.34 mmol/L, respectively). The index patient’s parents were not available for genetic testing.

Two patients were homozygous for a pathogenic *LIPA* variant, p.Gln298Gln. One patient, a female, 35 years of age at baseline with the highest documented LDL-C of 7.47 mmol/L, had no hepatomegaly or splenomegaly but did have small calcinates in the liver parenchyma on the computed tomography (CT) scan. Her alanine transaminase increased from 47 U/L to 58 U/L (reference < 31 U/L for women), and LDL-C decreased to 2.63 mmol/L with rosuvastatin 40 mg. She also had mild bilirubin elevations after the initiation of statin. Her Lp(a) level was 8.13 mg/dL. Another patient was a 38-year-old female at baseline with the highest documented LDL-C of 9.42 mmol/L, alanine transaminase of 55 U/L, and aspartate transaminase of 43 U/L. Her LDL-C decreased to 2.73 mmol/L with rosuvastatin 30 mg and 2.24 mmol/L with rosuvastatin 20 mg in combination with ezetimibe 10 mg. The levels of alanine transaminase were 49 U/L and 33 U/L, and the levels of aspartate transaminase were 44 U/L and 33 U/L (reference < 31 U/L for women) with these therapies, respectively. Her bilirubin levels were mildly elevated when measured after the initiation of statin. She had a history of fatty liver and one small hepatic calcinate without hepatomegaly described four years earlier on a CT scan. Her Lp(a) was 25.72 mg/dL.

Lysosomal acid lipase deficiency caused by homozygous *LIPA* variants may have a variable clinical course, and in the later-onset subtype, it is commonly misdiagnosed as FH or a hepatic disease, such as metabolic dysfunction-associated steatotic liver disease [15]. The onset of clinical manifestations has been described in age > 40 years in 4–5% of cases; hence, it is possible that these two patients may have the disease at age 35–38 years [16].

### 2.3. The Non-Monogenic Hypercholesterolemia Patients Have Elevated PRS Values

For comparison of the PRS values, we defined three groups: FH patients (n = 73, with phenocopy patients excluded), patients without the P/LP variants as non-FH (n = 259), and control group (n = 515). The high PRS was defined as the genetic score value above P_95_. A summary of all genetic findings in the FH, phenocopy patients, and non-FH patient groups is described in Table 3.

The median LDL-C PRS value was significantly higher in both the FH and non-FH groups compared to the controls (Figure 1A,B). Moreover, the non-FH group showed higher LDL-C PRS values than the FH group. We observed an overall good discriminatory ability between non-FH and controls (area under the receiver–operator characteristic curve (AUC) = 0.711) compared to poor discriminatory ability between the FH and control group (AUC = 0.620).

Individuals with the LDL-C PRS > P_95_ threshold were highly represented in both the FH and non-FH groups, accounting for 12.3% (n = 9) and 20.1% (n = 52) of their respective group sizes (Appendix A). However, the proportion of FH decreased in the higher PRS quartiles, with only 15.2% of the FH patients present in Q4 compared to 34.6% of patients in the lowest quartile (Q1) (Figure 2A). The chi-square test confirmed a significant difference in the FH patient distribution between the LDL-C PRS quartiles (*p* = 0.030), but pairwise comparisons did not show significance after adjusting for multiple tests.

In contrast, the median LPA PRS value did not differ significantly between FH versus non-FH and controls, although the median value in the non-FH group was markedly higher than in controls (Figure 1C,D). The PRS failed to discriminate both the FH and non-FH groups from controls (AUC = 0.585 and 0.558, respectively). Similar to LDL-C PRS, the highest percentage of non-FH individuals (8.9%) was in a subgroup of LPA PRS above the LPA PRS P_95_ (Appendix A). The proportion of FH cases was highest in Q3 (Figure 2B), while other quartiles showed a similar percentage. The distribution of FH patients did not differ significantly between the LPA PRS quartiles (*p* = 0.474).

Notably, the non-FH individuals were absent from the lowest LDL-C PRS percentile, while the FH individuals were absent from the 45–50th LDL-C PRS and lowest LPA PRS percentile (Appendix A).

### 2.4. Utility of PRS in Discriminating Priority VUS

In the studied cohort, 57 unique variants of uncertain significance (VUS) were found in 82 patients, and 6 patients had 2 such variants (Appendix A). Of these variants, 44 were missense, 8 were synonymous, 3 were splice site SNPs, and the remaining 2 were non-frameshift deletions. We have highlighted variants that were present in the classic FH genes and could be promoted to the LP class by obtaining one more moderate or supportive American College of Medical Genetics and Genomics/Association for Molecular Pathology (ACMG/AMP) criteria and defining them as high-priority VUS (Appendix A). In total, 11 unique VUS in 17 patients were applicable for high-priority VUS definition.

We further investigated the potential role of PRS in discriminating the priority variants from other VUS. The percentage of patients with high-priority VUS was higher in the Q1 and Q2 LDL-C PRS (Appendix A) and in the Q3 LPA PRS quartile (Appendix A). Nevertheless, the difference in the distribution of VUS variants across both the LDL-C and LPA PRS quartiles was not significant (*p* = 0.086 and 0.459, respectively), despite the significant difference in the VUS distribution between LDL-C PRS below versus above 50th percentiles (*p* = 0.031).

## 3. Discussion

### 3.1. Genetic Background of Clinical FH Patients

In this study, we continued to explore the genetic background of Latvian FH patients. In total, 21.5% of probands had a monogenic, FH-causing P or LP variation, a minor improvement in diagnostic rate compared to our previous work [7], though not significant (two-proportions z-test *p* = 0.955). Such a rate still places us at the lowest position of an estimated 20–30% diagnostic yield in Europe [9]. The potential explanation for a somewhat low prevalence of FH could be explained by several factors, as discussed in detail earlier [7]. Briefly, a part of the identified population-specific VUS in the major FH genes (*APOB*, *LDLR*, *LDLRAP1*, *PCSK9*) has the potential to be causative for the disease. For the selected high-priority VUS, cascade screening is planned to further accept (or exclude) these variants as explanations for a part of phenotypes in our cohort. Second, alternative causes of hyperlipidemia should be considered—four unique P and LP variants in other genes (*CYP27A1*, *ABCG5*, *ABCG8*, and *LIPA*) were found in seven patients (2% of all patients with positive monogenic variant), which increases the overall diagnostic yield to 23.6%. Third, the Latvian population is known to have higher cholesterol levels, which may lead to higher LDL-C levels irrespective of the genetic background [17]. However, patients without P/LP in our cohort had mean LDL-C levels of 6.8 mmol/L, which is significantly above the average LDL-C levels in our population. Interestingly, our patients with the P/LP FH variants had substantially higher mean LDL-C levels (8.3 mmol/L) than in the Danish population (5.40 and 6.09 mmol/L in the heterozygous *APOB* and *LDLR* carriers of the monogenic variant, respectively) [18].

Considering the low diagnostic yield, we expanded the search for a genetic explanation of the severe hypercholesterolemia phenotype in this study by investigating polygenic mechanisms and evaluating the PRS of LDL-C and Lp(a). High LDL-C PRS or/and LPA PRS were found in 26.3% of patients without P/LP or 20.1% of all patients, almost the same number of patients as those with the FH P/LP variants (21.5%). Thus, we were able to identify a genetic explanation of the FH phenotype in 43.7% of patients and not in the remaining majority.

Schwaninger and co-authors studied the prevalence of high genetic scores for LDL-C and Lp(a) among the FH patients in Austria. They reported a similar (22%) high PRS rate among patients without monogenic variants; however, they were mostly high LPA PRS cases (19.6% versus 6.8% in our sample), while in our study, 15.3% (vs. 4.3% in the Austrian group) had high LDL PRS (including double-hit high LDL-C and LPA PRS) [11]. Such differences could be due to different genetic backgrounds in the two populations. Notably, our sample had fewer cases of the FH monogenic variants (21.5% vs. 46.6% in the Austrian group).

A total of 57 VUS were found in 82 patients, and 17 were identified as priority variants with a higher potential for meeting the criteria for P/LP. The majority (n = 14 or 82.4%) of priority VUS were found in the subgroup of non-FH, and mostly (n = 12 or 70.6%) among individuals with non-high LDL-C PRS and LPA PRS. If these 12 VUS were to be confirmed as P or LP variants, they would contribute an additional 3.5% of genetic explanation for high cholesterol, making the total diagnostic yield 47.8% (Table 2). We were partly able to differentiate priority variants from other VUS using LDL-C PRS. To further prove the utility of PRS in selecting the variants of interest, more studies on uncertain variants are needed with cascade screening and functional tests included.

Two homozygous patients were found with the pathogenic *LIPA* variant, described previously as the most prevalent cause for lysosomal acid lipase deficiency (LAL-D), which could mimic FH and, thus, result in undertreatment and progressive liver failure [19]. Neither of the two patients had overt signs of LAL-D but had signs suggestive of the disease. Both patients experienced substantial reductions in LDL-C with statins while having mild alanine transaminase, aspartate transaminase, and bilirubin elevations. Their Lp(a) levels were <30.0 mg/dL and could not explain the cholesterol levels. One patient had LDL-C PRS within >P_95_. The lysosomal acid lipase activity had not yet been measured but was planned as a result of the genetic findings. The genetic variants in *LIPA* may have likely contributed to the high LDL-C levels.

The pathogenic/likely pathogenic variant of *CYP27A1* c.379C>T (p.Arg127Trp) was found in a heterozygous state in one female patient, aged 60 years, with no clinical signs characteristic of cerebrotendinous xanthomatosis that have been described in homozygous carriers of this variant and compound heterozygous cases [20,21]. Our patient’s highest LDL-C level was 7.92 mmol/L, and she had no signs of tendon xanthomas. We cannot exclude that this variant has contributed to the elevated LDL-C levels as there were no other alternative explanations (no VUS, Lp(a) was 24.9 mg/dL, LDL-C PRS < P_25_, and LPA PRS < P_95_).

Genetic variants in *ABCG5/G8* may result in sitosterolemia, which is an autosomal recessive disease, while the variants detected in our study are non-compound heterozygous: one patient had a frameshift insertion in *ABCG5* (probable FH), and three patients had the same SNP in *ABCG8* (two with probable and one with possible FH). However, multiple studies report patients with elevated LDL-C and CAD risk carrying heterozygous variants in either *ABCG5* or *ABCG8* [22]. This suggests that, despite being heterozygous, these variants may lead to a phenotype that mimics FH, and such discovery emphasizes the importance of additional lipid measurements for more precise diagnosis and treatment.

### 3.2. Study Strengths and Limitations

We performed genome-wide sequencing in a Latvian hypercholesterolemia patient population, which, compared to a small targeted panel, allowed us to investigate variants in both FH-causal and phenocopy genes. Furthermore, this approach allowed us to utilize the WGS data to calculate not only LDL-C PRS but also LPA PRS, as well as to potentially use the WGS data in further analyses. However, there are several limitations in our study. Our samples were selected from LRFH; therefore, our study focuses on individuals of almost exclusively Caucasian origin. Our patient sample may have a selection bias as we studied severely hypercholesterolemic patients referred to the Lipid Clinic all over Latvia. Some patients may have referral bias due to premature CAD caused by unfavorable lifestyle factors, especially if their LDL-C values are only slightly higher than 5.0 mmol/L. While our study was conducted on a Latvian cohort, we acknowledge that both overall [23] and FH-related genetic features [2,24] may vary by ethnic groups, and the patterns should be replicated in other populations.

While the DLCN criteria are valuable for identifying FH, they may have limitations. In younger individuals who may not yet exhibit clinical symptoms like tendon xanthomas or CAD, the FH diagnosis might be underestimated [25]. Conversely, in a high-risk country like Latvia, premature cardiovascular disease and elevated LDL-C levels could be attributed to lifestyle and other modifiable risk factors, potentially overestimating the likelihood of FH [17,26]. These considerations highlight the need for universal pediatric screening to enhance the diagnostic accuracy and identify the FH cases early [27].

While we did not conduct standardized lipid measurements at the time of inclusion, accessing historical laboratory reports allowed us to estimate the highest documented lipid levels and identify more cases with pretreatment levels [28]. However, the reported peak LDL-C levels may overestimate the average levels due to the regression to the mean phenomenon [29]. A minority of patients (n = 31, 9.1%) were on lipid-lowering therapy when maximal LDL-C was recorded, and we did not correct the values for the treatment effect. We did not gather information on what LDL-C measurement method was used in each of the several laboratory reports. However, since 2015, the direct LDL-C measurement method has been used in the majority of Latvian laboratories. Patients with greater TG levels and/or low LDL-C levels are more likely to have differences between different indirect techniques and direct LDL-C readings, whereas those with higher LDL-C levels are less likely to have these differences [30]. Therefore, we do not anticipate significant bias from the LDL-C measuring techniques in our group of patients with high LDL-C levels. Unfortunately, not all patients had recorded Lp(a), HDL-C, and TG levels in the Registry database. Finally, we focused on SNP and short indel discovery and did not perform a detailed investigation of SVs within our hypercholesterolemia population, which could contribute to overall diagnostic yield in our cohort. We plan to perform detailed CNV screening in the future, alongside cascade screening.

## 4. Methods

### 4.1. Study Cohort

The ongoing LRFH was established in 2015, and its general inclusion criteria and characteristics have been described previously [6]. The registry only includes adult patients (age > 18 years).

For this study, we selected index cases with LDL-C levels more than 5.0 mmol/L or total cholesterol more than 7.0 mmol/L who were also included in the LGDB from 2016 to 2023. We excluded relatives of index patients as well as cases of suspected secondary hypercholesterolemia (e.g., specific therapies, nephrotic syndrome, hypothyroidism).

Participant recruitment to LGDB, blood sampling, and associated data collection were performed according to standard procedures [31]. In total, 372 index patients with definite, probable, and possible FH diagnosis (based on DLCN criteria without prior genetic testing) were selected for WGS [32]. Additionally, one patient had two family members enrolled in the LRFH, and we decided to include them for monogenic variation screening via WGS. Lipid levels were obtained from all previous laboratory reports available at the time of patient inclusion in the registry. Biochemical measurements (as LDL-C, triglycerides, HDL-C, etc.) were measured by standard state laboratory assays. The highest documented LDL-C levels were reported, along with the corresponding TG and HDL-C levels at the time of the peak LDL-C measurement [28].

For PRS, a control group (n = 515) was selected from individuals within the LGDB resources with available WGS data. These individuals were recruited in LGDB for the Latvian Population Genome Reference project within the framework of the European “1+ Million Genome” initiative. This cohort comprises volunteer Latvian adult citizens, selected to ensure an even distribution of gender, nationality, and place of birth. To maintain genetic variability, participants with first-degree familial relationships were excluded, ensuring no two or more immediate relatives were included. Importantly, we deliberately avoided disease- or phenotype-based exclusion criteria to prevent introducing biases that could result in an artificially reduced or skewed PRS distribution for the selected traits. This approach allows the control group to retain its role as a genetically diverse and representative baseline for PRS evaluation.

### 4.2. Acquisition of Sequencing Data

The patient’s DNA was obtained from LGDB, which specializes in the processing and storage of biological samples (e.g., DNA, plasma, and sera). DNA was extracted from whole blood samples using the phenol–chloroform extraction method, followed by quality control in accordance with LGDB’s standard protocols, which comply with OECD and IARC guidelines, as described in detail previously [31]. The DNA libraries were prepared using the MGIEasy FS PCR-Free DNA Library Prep Set (MGI Tech Co., Ltd., Shenzhen, China) and the MGISP-960 High-throughput Automated Sample Preparation System (MGI Tech Co., Ltd., Shenzhen, China). Libraries were sequenced on the DNBSEQ-T10×4RS sequencing platform (MGI Tech Co., Ltd., Shenzhen, China) using the DNBSEQ-T10×4RS High-throughput Sequencing Set (FCL PE150) (MGI Tech Co., Ltd., Shenzhen, China), with at least 150 bp paired-end sequencing reads (30× sequencing coverage) per sample expected. The WGS was successful in 339 indexes and two family members with a mean sequencing depth of 37.14 (sd = 11). According to median values, 93% (IQR = 0.92–0.93) of WGS data reached at least 10× coverage, 68% (IQR = 0.56–0.78) reached at least 30× coverage, and 14% (IQR = 0.07–0.27) reached at least 50× coverage.

### 4.3. Variant Calling

The WGS data were trimmed using Trim Galore v.0.6.7 [33] and further processed with Nextflow-based pipeline nf-core/sarek v.3.3.2 [34], which utilizes Genomic Analysis Toolkit (GATK) 4 [35] best practices workflows. As part of the pipeline, the BWA-mem was selected for sequence alignment to the GATK.GRCh38 human reference genome; the GATK MarkDuplicates, GATK BaseRecalibrator, and GATK ApplyBQSR were selected for BAM file processing; the GATK HaplotypeCaller was selected for variant calling with nf-core/sarek option “--joint_germline” enabled for Genomic Variant Call Format (GVCF) file generation. The GVCF files will be further used for PRS evaluation.

### 4.4. Selection of Genes of Interest and Annotation

For this study, we analyzed genetic variants in four “standard” FH-genes (*LDLR*, *APOB*, *PCSK9*, *LDLRAP1)* and six additional genes from broader dyslipidemia gene panel (*ABCG5*, *ABCG8*, *APOE*, *CYP27A1*, *LIPA*, and *LPA*), described as FH-phenocopy genes [36,37]. The vcftools v.0.1.17 [38] was used for variant selection within genes of interest from multisample VCF (with FH patients only), and bcftools v.1.10.2 [39] was used for multiallelic call normalization. Variants were annotated with ANNOVAR version 2020-06-07 [40].

### 4.5. Variant Interpretation

The variants that had their gnomAD Genomes population maximum filtering allele frequency (PopMax FAF) > 0.5% (estimated FH prevalence) were filtered out and not interpreted, as they are expected to be too common in the context of FH.

Variants that were present in the ClinVar database [41] and (1) had classification “pathogenic” and/or “likely pathogenic”; (2) had two gold stars with at least 5 submitters or had three or more gold stars were assigned with the according germline classification. The remaining variants were interpreted according to ACMG/AMP guidelines [42], with The Clinical Genome Resource FH Variant Curation Expert Panel recommendations taken into account [43].

### 4.6. The Polygenic Score Calculation

For PRS calculation, the published validated PRS scores from The Polygenic Score (PGS) Catalog [44,45] for LDL-C (PGS Catalog ID PGS000115 [10]) and Lp(a) (PGS Catalog ID PGS000667 [46]) were obtained.

The FH group GVCFs were merged with the control group using GATK v.4.1.8.1 GenomicsDBImport tool within the intervals included in PRS scores. The joint genotyping was performed using GATK v.4.1.8.1 GenotypeGVCFs tool (with the option “--include-non-variant-sites” enabled).

The PRS values were calculated from the multisample VCF file using pgscatalog/pgsc_calc v2.0.0-alpha.5 pipeline [44].

### 4.7. Statistical Analysis and Software

Statistical analysis was performed using R Statistical Software (version 4.3.1 [47]) and SPSS (version 29.0.0.0; IBM Corporation, New York, NY, USA).

The one-way ANOVA test (with Tukey pairwise mean comparison post hoc test) was used for normally distributed continuous variables, and the pairwise Kruskall–Wallis test (with Dunn’s post hoc test) was used for non-normally distributed data. Proportions of categorical and ordinal variables were compared with the Chi-square test or Fisher’s exact test, as appropriate. The PRS percentiles and quartiles for the FH, non-FH, priority, and non-priority VUS groups were calculated based on control group PRS values. A significance level of *p* < 0.05 was chosen.

The WGS data analysis was performed on the Riga Technical University High-Performance Computing cluster. The GNU Parallel v.20200122 [48] was used to improve computational workflow efficiency. The Singularity v3.11.4 (Sylabs Inc., Reno, NV, USA) was used to install the necessary software and the Nextflow v.23.04 (Seqera Labs, Barcelona, Spain) for task automatization.

## 5. Conclusions

In this study that represents the first comprehensive genetic investigation of severe hypercholesterolemia in the Latvian patients with suspected or clinically diagnosed FH, a combination of monogenic FH, phenocopy variants, as well as LDL-C and LPA PRS genetically explained 43.7% of cases. Importantly, we identified six novel variants, contributing to the exploration of the genetic background of FH.

While monogenic variants were more frequent in patients with lower LDL-C PRS, high LDL-C PRS does not preclude monogenic FH, underscoring the necessity of comprehensive genetic testing for accurate diagnosis.

## Figures and Tables

**Figure 1 ijms-25-13466-f001:**
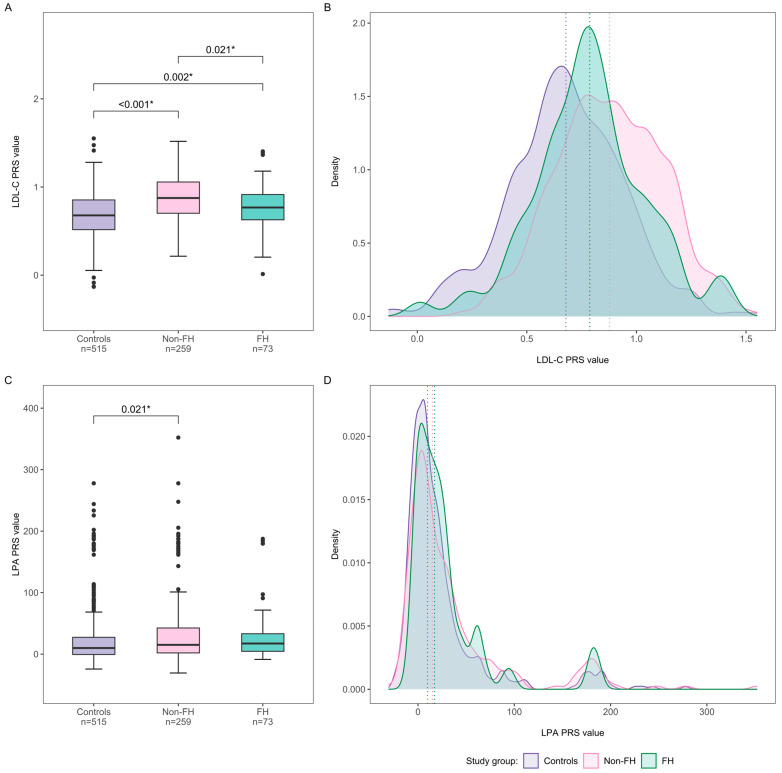
Distribution of LDL-C and LPA PRS values in FH (green) and non-FH (pink) patients vs. control group (purple). The box plots show the median, interquartile range, and minimum to maximum value of (**A**) LDL-C PRS values and (**C**) LPA PRS values in controls and hypercholesterolemia groups. Significant *p*-values are shown above the box plots. The density plots show the (**B**) LDL-C PRS and (**D**) LPA-C PRS distribution between controls and hypercholesterolemia groups. The dashed lines represent each group’s mean PRS value. LDL-C—low-density lipoprotein cholesterol; LPA—gene encoding lipoprotein (a); FH—familial hypercholesterolemia; Non-FH—FH patients without FH P/LP variants; PRS—polygenic risk score. *—statistically significant at *p* < 0.05.

**Figure 2 ijms-25-13466-f002:**
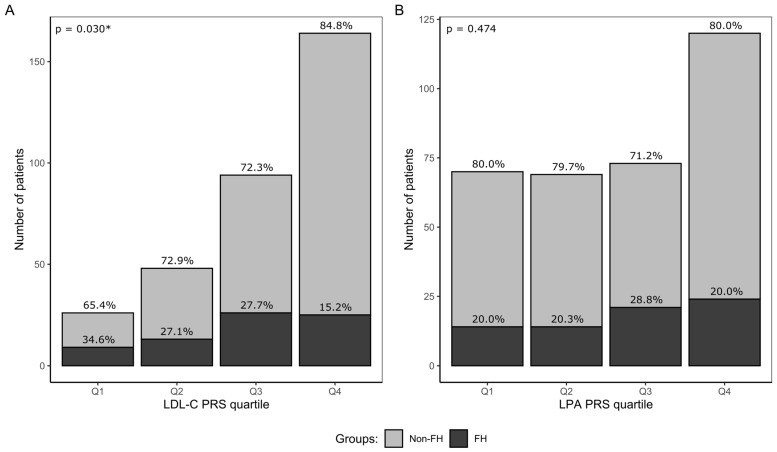
The number of hypercholesterolemia patients with (black bars) and without (gray bars) proven FH P/LP variants in (**A**) LDL-C PRS quartiles and (**B**) LPA PRS quartiles. The proportion of FH and non-FH groups is shown above bars as a percentage. Q1—<P_25_; Q2—P_25_–<P_50_; Q3—P_50_–<P_75_; Q4—≥P_75_; LDL-C—low-density lipoprotein cholesterol; LPA—gene encoding lipoprotein (a); FH—familial hypercholesterolemia; Non-FH—patients without P/LP variants; PRS—polygenic risk score. *—statistically significant at *p* < 0.05. Chi-square test *p*-values for LDL-C and LPA PRS are shown in the top-left corner.

**Table 1 ijms-25-13466-t001:** Characteristics of the clinical FH patient population.

Characteristic	FH Variants (P/LP)	Phenocopy Variants (P/LP)	Non-FH Cases	*p* Value *
Number of cases	n = 73	n = 7	n = 259	
Male, n (%)	26 (35.6)	2 (28.6)	104 (40.2)	0.483
Age				
Mean, SD	46.0 ± 12.9	51.1 ± 11.9	53.0 ± 11.4	<0.001 **
Median, IQR	46.0 [35.0–57.0]	57.0 [38.0–60.0]	53.0 [46.0–61.0]	<0.001 **
Highest documented				
LDL-C (mmol/L)				
Mean (SD)	8.3 ± 1.8	7.1 ± 1.4	6.8 ± 1.2	<0.001 **
Median (IQR)	8.1 [7.1–9.3]	6.8 [6.3–7.9]	6.7 [6.1–7.3]	<0.001 **
TG (mmol/L)	(n = 67)	(n = 6)	(n = 240)	
Median (IQR)	1.4 [1.0–1.9]	1.3 [1.1–1.5]	1.8 [1.3–2.6]	<0.001 **
HDL-C (mmol/L)	(n = 67)	(n = 6)	(n = 234)	
Mean (SD)	1.5 ± 0.4	1.5 ± 0.4	1.5 ± 0.4	0.227
Median (IQR)	1.4 [1.2–1.8]	1.6 × [1.3–1.8]	1.5 [1.3–1.8]	0.617
Lp(a) (mg/dL)	(n = 26)	(n = 2)	(n = 89)	
Median (IQR)	16.8 [8.6–86.1]	16.9 [-]	21.3 [10.4–79.8]	0.623
DLCN criteria, median (IQR) ***	9 [6–12]	7 [5–9]	6 [5–7]	<0.001 **
Clinical FH diagnosis ***				<0.001 **
Possible FH, n (%)	9 (12.3%)	2 (28.6%)	88 (34.0%)	
Probable FH, n (%)	27 (37.0%)	3 (42.8%)	123 (47.5%)	
Definite FH, n (%)	37 (50.7%)	2 (28.6%)	48 (18.5%)	
Tendon xanthomas, n (%)	18 (24.7%)	1 (14.3%)	36 (13.9%)	0.028 **
Premature arcus cornealis < 45 years, n (%)	5 (6.8%)	0 (0%)	5 (1.9%)	0.045 **
CAD, n (%)	37 (50.7%)	3 (42.9%)	136 (52.5%)	0.783
Premature CAD, n (%)	32 (43.8%)	3 (42.9%)	88 (34.0%)	0.121
Myocardial infarction, n (%)	6 (8.2%)	1 (14.3%)	26 (10.0%)	0.642
PCI, n (%)	10 (13.7%)	2 (28.6%)	41 (15.8%)	0.638
CABG, n (%)	3 (4.1%)	0 (0%)	5 (1.9%)	0.382
Diabetes, n (%)	1 (1.4%)	0 (0%)	13 (5.0%)	0.319
Hypertension, n (%)	22 (30.1%)	2 (28.6%)	124 (47.9%)	0.007 **
Smoking, n (%) ****	28 (38.4%)	2 (28.6%)	87 (33.6%)	0.450
LDL-C PRS > 95%, n (%)	9 (12.3%)	1 (14.3%)	52 (20.1%)	0.131
LPA PRS > 95%, n (%)	4 (5.5%)	1 (14.3%)	23 (8.9%)	0.348
LDL-C PRS < 5%, n (%)	2 (2.7%)	0 (0%)	0 (0%)	0.048 **
LPA PRS < 5%, n (%)	0 (0%)	0 (0%)	15 (5.8%)	0.049 **

FH—familial hypercholesterolemia, SD—standard deviation, IQR—interquartile range, HDL-C—high-density lipoprotein cholesterol (reference ≥ 1.0 for men and ≥1.2 for women), LDL-C—low-density lipoprotein cholesterol (reference < 3.0 mmol/L), DLCN—Dutch Lipid Clinic Network, CAD—coronary artery disease, PCI—percutaneous coronary intervention, CABG—coronary artery bypass grafting, PRS—polygenic risk score, Lp(a)—lipoprotein (a) (reference < 30 mg/dL), LPA—gene encoding lipoprotein (a), LP—likely pathogenic variant, P—pathogenic variant, TG—triglycerides (reference < 1.7 mmol/L). *—FH vs. non-FH cases. **—statistically significant at *p* < 0.05. ***—estimated before genetic data were known. ****—current or previous smoking.

**Table 2 ijms-25-13466-t002:** Pathogenic and/or likely pathogenic variants found in Latvian FH cohort.

Disease	Gene	Variant	rsID	ClinVar ID	ClinVar Stars	Zygosity, VEP Consequence	Clinical Significance	Cases (n)
FH	*APOB*	c.10579C>G (p.Arg3527Gly)	NA	NA	NA	Het, missense variant	LP ^2^	1
		c.10580G>A (p.Arg3527Gln) ^1^	rs5742904	17890	2	Het, missense variant	P/LP	13
	*LDLR*	c.11G>A (p.Trp4Ter) ^1^	rs201016593	250973	2	Het, stop gained	P	1
		c.214del (p.Asp72ThrfsTer134)	rs879254438	251078	2	Het, frameshift deletion	P	1
		c.397G>A (p.Asp133Asn)	rs879254513	251202	1	Het, missense variant	LP ^2^	1
		c.427T>A (p.Cys143Ser) ^1^	rs875989901	920596	2	Het, missense variant	P ^2^	1
		c.463T>G (p.Cys155Gly)	rs879254535	251239	2	Het, missense variant	P/LP	1
		c.504C>A (Asp168Glu)	rs777321035	251261	1	Het, missense variant	P ^2^	1
		c.530C>T (p.Ser177Leu) ^1^	rs121908026	3686	3	Het, missense variant	P	2
		c.616A>C (p.Ser206Arg)	rs2077277985	869389	1	Het, missense variant	P ^2^	1
		c.625T>G (p.Cys209Gly) ^1^	rs1600711065	684864	1	Het, missense variant	LP ^2^	1
		c.651TGG[1] (p.Gly219del)	rs121908027	226329	3	Het, nonframeshift deletion	P	3
		c.666C>A (p.Cys222Ter) ^1^	rs756613387	251364	2	Het, stop gained	P	1
		c.682G>C (p.Glu228Gln)	rs121908029	251393	2	Het, missense variant	P/LP	2
		c.761A>C (p.Gln254Pro)	rs879254667	251437	2	Het, missense variant	P/LP	1
		c.798T>A (p.Asp266Glu) ^1^	rs139043155	161287	3	Het, missense variant	P	1
		c.888C>A (p.Cys296Ter)	rs879254708	251504	2	Het, stop gained	P ^2^	1
		c.910G>A (p.Asp304Asn) ^1^	rs121908030	3692	3	Het, missense variant	P	2
		c.986G>A (p.Cys329Tyr) ^1^	rs761954844	226344	2	Het, missense variant	P/LP	8
FH	*LDLR*	c.1061-2A>C	NA	NA	NA	Het, splice site variant	LP ^2^	1
		c.1187-10G>A	rs765696008	226349	2	Het, splice site variant	P/LP	1
		c.1202T>A (p.Leu401His)	rs121908038	3735	1	Het, missense variant	LP ^2^	2
		c.1222G>A (p.Glu408Lys) ^1^	rs137943601	36453	3	Het, missense variant	LP	2
		c.1222_1223insAGGTC (p.Lys411SerfsTer4)	NA	NA	NA	Het, frameshift insertion	P ^2^	1
		c.1277T>C (p.Leu426Pro)	rs879254851	251763	2	Het, missense variant	P/LP	2
		c.1285G>A (p.Val429Met) ^1^	rs28942078	3694	3	Het, missense variant	P	2
		c.1292C>T (p.Ala431Val) ^1^	NA	NA	NA	Het, missense variant	LP ^2^	2
		c.1618G>A (p.Ala540Thr)	rs769370816	226363	3	Het, missense variant	P	3
		c.1730G>C (p.Trp577Ser)	rs138947766	252003	2	Het, missense variant	P/LP	1
		c.1756T>C (Ser586Pro)	rs2147257524	1334395	1	Het, missense variant	LP ^2^	1
		c.1775G>A (p.Gly592Glu) ^1^	rs137929307	161271	3	Het, missense variant	P	5
		c.1784G>A (p.Arg595Gln)	rs201102492	183126	3	Het, missense variant	P	1
		c.1846-2A>G	NA	NA	NA	Het, splice site variant	LP ^2^	1
		c.1978C>T (p.Gln660Ter) ^1^	rs193922569	36458	2	Het, stop gained variant	P ^2^	1
		c.1998G>A(p.Trp666Ter) ^1^	rs752935814	252161	2	Het, stop gained	P	3
		c.2030G>A (p.Cys677Tyr)	rs875989938	252178	2	Het, missense variant	LP ^2^	1
		c.2045T>C (p.Leu682Pro)	rs879255119	252189	1	Hom, missense variant	LP ^2^	1
		c.2180_2184dup (p.Leu729SerfsTer3)	rs1555808044	403646	2	Het, frameshift duplication	P ^2^	1
		c.2416dup (p.Val806GlyfsTer11)	rs773618064	252330	3	Het, frameshift duplication	P	1
Other	*ABCG5*	c.449_450insN[109] (p.Thr151LeufsTer83)	NA	NA	NA	Het, frameshift insertion	P ^2^	1
	*ABCG8*	c.1083G>A (p.Trp361Ter)	rs137852987	4967	2	Het, stop gained	P/LP	3
	*CYP27A1*	c.379C>T:p.Arg127Trp	rs201114717	65865	2	Het, missense variant	P/LP	1
	*LIPA*	c.894G>A (p.Gln298=)	rs116928232	203361	2	Hom, synonymous variant	P/LP	2

FH—familial hypercholesterolemia; VEP—Variant Effect Predictor; Het—heterozygous; Hom—homozygous; P—pathogenic; LP—likely pathogenic; ^1^—variant was found in our previous study; ^2^—variant was classified according to ACMG/AMP guidelines.

**Table 3 ijms-25-13466-t003:** Summary of genetic findings in FH patients.

Patient Group	Number	Percentage of All Patients	High PRS	Number	Percentage of Subgroup	Percentage of All Patients	Explains Phenotype Among All Patients		VUS	
							n	%	n	%
FH P/LP	73	21.5%	LDL-C PRS	9	12.3%	2.6%	73	21.5%	0	0.0%
			LPA PRS	4	5.5%	1.2%			0	0.0%
			Both PRS	0	0	0			0	NA
			None	60	82.2%	17.7%			3	5.0%
Other P/LP	7	2.1%	LDL-C PRS	1	14.3%	0.3%	7	2.1%	0	0.0%
			LPA PRS	1	14.3%	0.3%			0	0.0%
			Both PRS	0	0	0			0	NA
			None	5	71.4%	1.5%			0	0.0%
No P/LP	259	76.4%	LDL-C PRS	45	17.4%	13.3%	68	20.1%	2	4.4%
			LPA PRS	16	6.2%	4.7%			0	0.0%
			Both PRS	7	2.7%	2.1%			0	0.0%
			None	191	73.7%	56.3%			12	6.3%
Total	339	100.0%		339		100	148	43.7%	17	

FH—familial hypercholesterolemia, LDL-C—low-density lipoprotein cholesterol, LPA—gene encoding lipoprotein (a), P—pathogenic variant, LP—likely pathogenic variant, VUS—variant of uncertain significance, PRS—polygenic risk score.

## Data Availability

The majority of the data generated and analyzed during this study are included in this published article and the corresponding Appendix A. The raw genotyping data are under restricted access from the Genome Database of Latvian Population and are available for research purposes on a reasonable request.

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
