# Peer review of "Improved Genetic Characterization of Hypercholesterolemia in Latvian Patients with Familial Hypercholesterolemia: A Combined Monogenic and Polygenic Approach Using Whole-Genome Sequencing"

_ijms, 2024, doi:10.3390/ijms252413466_

Round 1
Reviewer 1 Report
Comments and Suggestions for Authors
Atava and colleagues have performed whole genome sequencing in 339 clinically diagnosed familial hypercholesterolemia (FH) patients to identify those with monogenic and polygenic hypercholesterolemia.
The paper is generally well thought out and well written.
My major concern is that patients with ‘phenocopy’ variants are grouped together, and then grouped together with those with FH variants. I think it would be better if the phenocopies were excluded from analysis, or at least the two patients with lysosomal acid lipase deficiency, a monogenic disorder that is quite different to FH. The others can be reassigned or excluded. Inclusion of a patient who is a heterozygous carrier of a CYP27A1 pathogenic variant in the ‘monogenic hypercholesterolemia’ group is concerning, given they are asymptomatic and patients who have the recessive disorder cerebrotendinous xanthomatosis do not have elevated LDL-cholesterol levels – it is their xanthomas that might mis-diagnose them as FH. Similarly, while studies indicate a small excess of ABCG5 and ABCG8 heterozygotes among hypercholesterolemic patients, this does not mean they have the same phenotype as a patient with a heterozygous pathogenic LDLR variant. I recommend the group ‘monogenic hypercholesterolemia’ be simply the FH group and be reanalysed as such.
Other points
Table 1 – Needs statistics – indicate which groups are significantly different for which parameters; are Dutch scores, LDL, PRS different? Hypertension line is missing data. Include the available data for Lp(a) levels and indicate n.
Ensure units are provided throughout the manuscript – e.g. for ALT, AST, LDL. It would be helpful to provide reference interval / decision limit for parameters mentioned e.g. ALT, AST and Lp(a).
For discussion - regarding the two patients found to have homozygous LIPA variants – is it unusual for patients with this disorder to not have significant/overt disease in their late 30s?
Were any APOE p.Leu167del variants identified as a cause of FH, or APOE E2/2 homozygotes with high triglycerides that could be dysbetalipoproteinemia rather than FH?
Figure 2 – indicate any statistically significant differences between categories
Figure 3 – given the findings are not significant and are lower priority this should be moved to the supplementary.
Table 3 – one of the headings is ‘High PRS’ but it is not defined in the table or results section what that actually means, in plain English. Is this individuals with a score >95th population percentile, or something else?
The lack of copy number variant analysis means that potentially up to 10% of molecular diagnoses were missed. Would it be possible for even just those with the highest LDL-c without a detected pathogenic variant be tested with MLPA to look for large LDLR deletions/duplications?
Please comment on the cost and practicality of whole genome sequencing to diagnose FH, compared to a small targeted panel.
Author Response
We thank the reviewer for the careful reading of the manuscript and constructive remarks. We have taken the comments on board to improve and clarify the manuscript. Please find below a detailed point-by-point response to all comments.
Comments1: My major concern is that patients with ‘phenocopy’ variants are grouped together, and then grouped together with those with FH variants. I think it would be better if the phenocopies were excluded from analysis, or at least the two patients with lysosomal acid lipase deficiency, a monogenic disorder that is quite different to FH. The others can be reassigned or excluded. Inclusion of a patient who is a heterozygous carrier of a CYP27A1 pathogenic variant in the ‘monogenic hypercholesterolemia’ group is concerning, given they are asymptomatic and patients who have the recessive disorder cerebrotendinous xanthomatosis do not have elevated LDL-cholesterol levels – it is their xanthomas that might mis-diagnose them as FH. Similarly, while studies indicate a small excess of ABCG5 and ABCG8 heterozygotes among hypercholesterolemic patients, this does not mean they have the same phenotype as a patient with a heterozygous pathogenic LDLR variant. I recommend the group ‘monogenic hypercholesterolemia’ be simply the FH group and be reanalysed as such.
Response 1: We agree with the reviewer and we excluded the 7 patients with phenocopies from further analyses, and have updated the manuscript accordingly. We also removed the terms “monogenic” and “non-monogenic” hypercholesterolemia, and replaced them with FH and non-FH terms throughout the manuscript.
We consequently updated the Results 2.3. sub-section with appropriate AUC, p-values, and patient counts and proportions. We updated figure 1 and 2 accordingly to the new groups (FH and non-FH).
We did not, however, exclude patients with phenocopy variants from the manuscript part, where we explore the utility of PRS in discriminating priority VUS. We believe that proven monogenic cause for lipidemia (such as LIPA homozygous variants) does not exclude the possibility that a patient could carry additional monogenic variants linked to other lipid-related conditions. Moreover, these patients may also harbor a significant burden of common low-effect variants contributing to dyslipidemia, resulting in a high PRS.
Comments 2: Table 1 – Needs statistics – indicate which groups are significantly different for which parameters; are Dutch scores, LDL, PRS different?
Response 2: We have calculated p values for all the variables. Considering the small number of cases in the phenocopy group we decided to compare only FH vs non-FH cases. We hope the reviewer would agree with such an approach. We have also marked the statistically significant values (at p<0.05) with two asterixes.
Comments 3: Hypertension line is missing data.
Response3: The Hypertension line is complete now.
Comments 4: Include the available data for Lp(a) levels and indicate n.
Response 4: We have added Lp(a) data in Table 1. As previously already noted in discussion, not all patients had it measured. We have indicated the number of patient cases in each group with known Lp(a) levels.
Comments 5: Ensure units are provided throughout the manuscript – e.g. for ALT, AST, LDL.
Response 5: We have added the units where they were missing.
Comments 6: It would be helpful to provide reference interval / decision limit for parameters mentioned e.g. ALT, AST and Lp(a).
Response 6: We have indicated the following references:
Line 156, for alanine transaminase: “(reference <31 U/L for women)”
Line 163, for aspartate transaminase: “(reference <31 U/L for women)”
Table 1 footnote, for LDL-C: “(reference <3.0 mmol/L)”
Table 1 footnote, for Lp(a): “(reference <30 mg/dL)”
Table 1 footnote, for HDL-C: “(reference >1.0 for men and >1.2 for women)”
Table 1 footnote, for triglycerides: “(reference <1.7 mmol/L)”
Comments 7: For discussion - regarding the two patients found to have homozygous LIPA variants – is it unusual for patients with this disorder to not have significant/overt disease in their late 30s?
Response 7: We have added the following sentence in the discussion, lines 167-171.
“Lysosomal acid lipase deficiency caused by homozygous LIPA variants may have variable clinical course, and in later-onset subtype, it is commonly misdiagnosed as an FH or a hepatic disease such as metabolic dysfunction-associated steatotic liver disease [Reiner et al., 2014]. The onset of clinical manifestations has been described in age >40 years in 4-5% of cases, hence it is possible that these two patients may have the disease at age 35-38 years [Bernstein et al., 2013].”
References:
Reiner Z, et al. Lysosomal acid lipase deficiency e An under-recognized cause of dyslipidaemia and liver dysfunction. Atherosclerosis 235 (2014) 21-30
Bernstein DL, Hulkova H, Bialer MG, Desnick RJ. Cholesteryl ester storage disease: review of the findings in 135 reported patients with an underdiagnosed disease. J Hepatol 2013;58:1230-43.
Comments 8: Were any APOE p.Leu167del variants identified as a cause of FH, or APOE E2/2 homozygotes with high triglycerides that could be dysbetalipoproteinemia rather than FH?
Response 8: We performed the variant analysis within the APOE gene and did not find APOE p.Leu167del variant or any APOE homozygous variants. We did find APOE c.434G>A (p.Gly145Asp) in one patient, which presented in heterozygous form and was classified as VUS. This variant is shown in Table S2.
Comments 9: Figure 2 – indicate any statistically significant differences between categories
Response 9: We revised the following sentence in the Results (lines 192-194):
“The chi-square test confirmed a significant difference in FH patient distribution between LDL-C PRS quartiles (p = 0.030), but pairwise comparisons did not show significance after adjusting for multiple tests.”
We added the LDL-C and LPA-PRS Chi-square test p-values to Figure 2 and revised figure legend.
Although the Chi-square test shows significant difference in FH versus non-FH LDL-C PRS values, pairwise comparisons of PRS quartiles did not yield significant p-values after the Bonferroni correction, and we decided to not add these p-values to Figure 2.
Comments 10: Figure 3 – given the findings are not significant and are lower priority this should be moved to the supplementary.
Response 10: We agree with the reviewer and have implemented the suggested change. Figure 3 has been moved to the supplementary materials and is now labeled as Figure S2.
Comments 11: Table 3 – one of the headings is ‘High PRS’ but it is not defined in the table or results section what that actually means, in plain English. Is this individuals with a score >95th population percentile, or something else?
Response 11: We have added the following sentence in the Results, line 175:
“The high PRS was defined as the genetic score value above P95.”
Comments 12: The lack of copy number variant analysis means that potentially up to 10% of molecular diagnoses were missed. Would it be possible for even just those with the highest LDL-c without a detected pathogenic variant be tested with MLPA to look for large LDLR deletions/duplications?
Response 12: We agree with the reviewer and understand how important a detailed analysis of the CNV is. Unfortunately, at the moment we are not able to conduct MLPA in this study. We are aware of the shortcomings of our approach and mention this in the "Strengths and limitations" section, and plan to implement these methods in future, lines 337-338:
“We plan to perform detailed CNV screening in the future, alongside cascade screening.”
Comments 13: Please comment on the cost and practicality of whole genome sequencing to diagnose FH, compared to a small targeted panel.
Response13: We have updated the following sentences in the Discussion, commenting on WGS practicality, line 309-313:
“We performed genome-wide sequencing in a Latvian hypercholesterolemia patient population, which, compared to a small targeted panel, allowed us to investigate variants in both FH-causal and phenocopy genes. Furthermore, this approach allows us to utilize WGS data to calculate not only LDL-C PRS but also LPA PRS, as well as potentially use WGS data in further analyses.”
Reviewer 2 Report
Comments and Suggestions for Authors
The study might provide the fundamental information. Some points could be reconsidered and added for potential improvement of the quality of the paper.
1. How specific are the study results to Latvian patients with FH? The fact that the study was focused on ‘Latvian’ patients is also described in Title. The specificity of results to Latvian patients could be more discussed.
2. In relation to the above, clinical practice, screening, prognosis, and epidemiology of FH in Latvian people could be summarized in Introduction.
3. Are there any problems in using the DLCN criteria?
4. In Methods, the methods used to measure the LDL-C and/or lipid levels must be described.
5. The conclusion might be (a little bit) weak at the end of text. A sharp description of conclusion would be expected.
6. Line 40; the estimation of 1:200 may be overrated for heterozygous FH patients based on the various references. Could the estimated rate have some width with some references?
7. Line 40; the estimation of 1:200 may be overrated for heterozygous FH patients based on the various references. Could the estimated rate have some width with some references?
8. Line 41; the end of sentence might have some references.
9. Line 46; the end of sentence might have some references.
10. Table 1; could you please demonstrate the HDL-C levels?
11. Table 1; could you please demonstrate the triglyceride levels?
12. Table 1; could you please show the prevalence of smokers?
13. Line 139; the end of sentence should have some references for the definition of LDL-C levels.
14. Line 143, 146, 148, 149, 272; AST and ALT may be fully spelled out.
15. Line 298; how large was the selection bias? The speculated affect on the results could be more detailed.
16. Line 310; the (excluded or included) criteria in recruiting the patients should be introduced.
17. Line 323-324; the accuracy and reproducibility of sequencing data could be added.
18. Line 315; the DLCN criteria could be detailed with some references.
19. Line 408; was informed consent given in the ‘written or signed’ style? If so, that should be described clearly, especially in gene-related studies.
Author Response
We thank the reviewer for the careful reading of the manuscript and constructive remarks. We have taken the comments on board to improve and clarify the manuscript. Please find below a detailed point-by-point response to all comments.
Comments 1: How specific are the study results to Latvian patients with FH? The fact that the study was focused on ‘Latvian’ patients is also described in Title. The specificity of results to Latvian patients could be more discussed.
Response 1: We have added the following sentences in the discussion, lines 318-320:
“While our study was conducted on a Latvian cohort, we acknowledge that genetic features may vary by ethnic groups and the patterns should be replicated in other populations.”
Comments 2: In relation to the above, clinical practice, screening, prognosis, and epidemiology of FH in Latvian people could be summarized in Introduction.
Response 2: We have added the following paragraph in the Introduction, lines 48-55:
“Familial Hypercholesterolemia in Latvia is diagnosed primarily using the Dutch Lipid Clinic Network (DLCN) criteria through opportunistic screening of adults and cascade screening of first-degree relatives of index cases. Universal FH screening is not implemented. The Latvian Registry of FH (LRFH), established in 2015, aims to improve FH diagnosis and management. It focuses on identifying individuals, initiating early treatment, and implementing cascade screening. While genetic testing is not routinely available, genetic studies are conducted based on research grants, with many FH patients participating in the Genome Database of Latvian Population (LGDB).”
Comments 3: Are there any problems in using the DLCN criteria?
Response 3: We have added the following sentences in the Discussion, lines 321-327:
“While the DLCN criteria are a valuable tool for identifying FH, they may have limitations. In younger individuals who may not yet exhibit clinical symptoms like tendon xanthomas or CAD, the FH diagnosis might be underestimated. Conversely, in a high-risk country like Latvia, premature cardiovascular disease and elevated LDL-C levels could be attributed to lifestyle and other modifiable risk factors, potentially overestimating the likelihood of FH. These considerations highlight the need for universal pediatric screening to enhance diagnostic accuracy and identify FH cases early.”
Comments 4: In Methods, the methods used to measure the LDL-C and/or lipid levels must be described.
Response 4: We have added the following sentences.
In Methods, lines 353-356:
“Lipid levels were obtained from all previous laboratory reports available at the time of patient inclusion in the Registry. The highest documented LDL-C levels were reported, along with the corresponding TG and HDL-C levels at the time of the peak LDL-C measurement. ”
In Discussion, lines 328-332, we have added and revised several sentences which now stand as follows:
“While we did not conduct standardized lipid measurements at the time of inclusion, accessing historical laboratory reports allowed us to estimate the highest documented lipid levels and identify more cases with pretreatment levels. However, the reported peak LDL-C levels may overestimate the average levels due to the regression to the mean phenomenon.”
Comments 5: The conclusion might be (a little bit) weak at the end of text. A sharp description of conclusion would be expected.
Response 5: We revised and added the following sentences to the Conclusions, lines 435-442:
“In this study that represents the first comprehensive genetic investigation of severe hypercholesterolemia in the Latvian patients with suspected or clinically diagnosed FH, a combination of monogenic FH, phenocopy variants as well as LDL-C and LPA PRS genetically explained 43.7% of cases. Importantly, we identified 6 novel variants, contributing to the exploration of the genetic background of FH.
While monogenic variants were more frequent in patients with lower LDL-C PRS, high LDL-C PRS does not preclude monogenic FH, underscoring the necessity of comprehensive genetic testing for accurate diagnosis.”
Comments 6: Line 40; the estimation of 1:200 may be overrated for heterozygous FH patients based on the various references. Could the estimated rate have some width with some references?
Comments 7: Line 40; the estimation of 1:200 may be overrated for heterozygous FH patients based on the various references. Could the estimated rate have some width with some references?
Comments 8: Line 41; the end of sentence might have some references.
Response to Comments 6-8: We agree that the FH prevalence estimates vary in various reports and the initially written numbers did not reflect it. We have corrected the statements and added three references, lines 39-41:
“Familial hypercholesterolemia (FH) is one of the most common autosomal dominant genetic disorders (estimated prevalence of 1:200-313 and 1:160,000-300,000 for heterozygous (HeFH) and homozygous (HoFH) forms, respectively [1, we added two references: Beheshti et al and Cuchel et al.])”
1) Sabina O. Beheshti, Christian M. Madsen, Anette Varbo, Børge G. Nordestgaard, Worldwide Prevalence of Familial Hypercholesterolemia: Meta-Analyses of 11 Million Subjects, Journal of the American College of Cardiology, Volume 75, Issue 20, 2020, Pages 2553-2566, ISSN 0735-1097, https://doi.org/10.1016/j.jacc.2020.03.057.
2) Cuchel M, Bruckert E, Ginsberg HN, Raal FJ, Santos RD, Hegele RA, Kuivenhoven JA, Nordestgaard BG, Descamps OS, Steinhagen-Thiessen E, Tybjærg-Hansen A, Watts GF, Averna M, Boileau C, Borén J, Catapano AL, Defesche JC, Hovingh GK, Humphries SE, Kovanen PT, Masana L, Pajukanta P, Parhofer KG, Ray KK, Stalenhoef AF, Stroes E, Taskinen MR, Wiegman A, Wiklund O, Chapman MJ; European Atherosclerosis Society Consensus Panel on Familial Hypercholesterolaemia. Homozygous familial hypercholesterolaemia: new insights and guidance for clinicians to improve detection and clinical management. A position paper from the Consensus Panel on Familial Hypercholesterolaemia of the European Atherosclerosis Society. Eur Heart J. 2014 Aug 21;35(32):2146-57. doi: 10.1093/eurheartj/ehu274. Epub 2014 Jul 22. PMID: 25053660; PMCID: PMC4139706.10.1093/eurheartj/ehu274
Comments 9: Line 46; the end of sentence might have some references.
Response 9: We have added a reference to this sentence (now line 46):
”The loss of function (LoF) variants in LDLR (most prevalent), APOB, and gain of function variants in PCSK9 genes and, rarely, homozygous LoF variations in LDLRAP1 gene result in FH phenotype, presenting in two forms [Sharifi et al]”
Sharifi M, Futema M, Nair D, Humphries SE. Genetic Architecture of Familial Hypercholesterolaemia. Curr Cardiol Rep. 2017 May;19(5):44. doi: 10.1007/s11886-017-0848-8. PMID: 28405938; PMCID: PMC5389990.
Comments 10: Table 1; could you please demonstrate the HDL-C levels?
Comments 11: Table 1; could you please demonstrate the triglyceride levels?
Response to Comments 10-11: We have now given the numbers of patients with known values and their statistics in Table 1. Unfortunately, not all patients had their HDL-C and TG levels recorded at the baseline. We therefore have added the following sentence in Discussion, lines 333-334:
“Unfortunately, not all patients had recorded Lp(a), HDL-C and TG levels in the Registry database.”
Comments 12: Table 1; could you please show the prevalence of smokers?
Response 12: The Table 1 is updated with data about smokers.
Comments 13: Line 139; the end of sentence should have some references for the definition of LDL-C levels.
Response 13: We are unsure if we fully understand the comment. While the laboratory reference value for LDL-C is <3.0 mmol/L, respectfully, in our view it is irrelevant in this context as all patients included in the study, by definition, had significantly elevated LDL-C levels far exceeding the reference range. Given that we have included the LDL-C reference value in Table 1 footnote, we hope the reviewer would agree that mentioning it in the main text here is unnecessary.
Comments 14: Line 143, 146, 148, 149, 272; AST and ALT may be fully spelled out.
Response 14: Now AST and ALT are fully spelled as aspartate transaminase and alanine transaminase.
Comments 15: Line 298; how large was the selection bias? The speculated affect on the results could be more detailed.
Response 15: We added the following sentence (lines 316-318):
“Some patients may have referral bias due to premature CAD caused by unfavorable lifestyle factors, especially if their LDL-C values are only slightly higher than 5.0 mmol/L.”
Comments 16: Line 310; the (excluded or included) criteria in recruiting the patients should be introduced.
Response 16: We added the following sentences (now lines 342-347):
“The Registry only includes adult patients (age >18 years).
For this study, we selected index cases with LDL-C levels more than 5.0 mmol/L or total cholesterol more than 7.0 mmol/L, who were also included in the LGDB from 2016 to 2023. We excluded relatives of index patients as well as cases of suspected secondary hypercholesterolemia (e.g., specific therapies, nephrotic syndrome, hypothyroidism).”
Comments 17: Line 323-324; the accuracy and reproducibility of sequencing data could be added.
Response 17: We added the following sentence to Methods, describing sequencing metrics (lines 381-383):
“According to median values, 93% (IQR = 0.92-0.93) of WGS data reached at least 10X coverage, 68% (IQR = 0.56-0.78) reached at least 30X coverage, and 14% (IQR = 0.07-0.27) reached at least 50X coverage.”
Comments 18: Line 315; the DLCN criteria could be detailed with some references.
Response 18: We have added the following reference (now line 351):
Nordestgaard BG, Chapman MJ, Humphries SE, Ginsberg HN, Masana L, Descamps OS, Wiklund O, Hegele RA, Raal FJ, Defesche JC, Wiegman A, Santos RD, Watts GF, Parhofer KG, Hovingh GK, Kovanen PT, Boileau C, Averna M, Borén J, Bruckert E, Catapano AL, Kuivenhoven JA, Pajukanta P, Ray K, Stalenhoef AF, Stroes E, Taskinen MR, Tybjærg-Hansen A; European Atherosclerosis Society Consensus Panel. Familial hypercholesterolaemia is underdiagnosed and undertreated in the general population: guidance for clinicians to prevent coronary heart disease: consensus statement of the European Atherosclerosis Society. Eur Heart J. 2013 Dec;34(45):3478-90a. doi: 10.1093/eurheartj/eht273. Epub 2013 Aug 15.
Comments 19: Line 408; was informed consent given in the ‘written or signed’ style? If so, that should be described clearly, especially in gene-related studies.
Response 19: We have revised the sentences to the following (lines 462-463): “Two separate written and signed informed consents for participation in the LRF and the LGDB were obtained from all subjects involved in the study.”
Reviewer 3 Report
Comments and Suggestions for Authors
The authors present an interesting article in which an approach is developed and scrutinised to determine the risk of hypercholesterolemia in patient populations based on the absence/presence of particular alleles that are associated with such. In essence, this approach aims to compare and contrast existing diagnosis/presence of conditions associated with hypercholesterolemia with the absence/presence of monogenic and polygenic makeups in an effort to determine what weight particular alleles carry as a predictive index. In comparing a control group to patient groups, the authors reveal a number of predictive combinations and variants of alleles in the patient populations that could be used effectively for determining familial risk.
In reviewing the manuscript, I made a couple of observations. The following should be considered by the authors when preparing a suitable revision.
1. For the control group that is mentioned in the methods, inclusion/exclusion criteria for designating them as the control group should be given in Section 4.1.
2. For the blood samples that were utilised as part of the study, more details on the samples and their storage should be given. For example, how long did it take to acquire the samples for the entire cohort, or were these samples that were already taken and the authors granted access to? What was the average age of the samples in terms of storage times? Was the DNA extracted immediately or were the samples stored for a time before the DNA was extracted? Details such as these in terms of the consistency of sampling would be useful.
3. The authors refer to an article which describes how the DNA was acquired from blood samples that were taken from participants – can the authors confirm that they carried out the same? In other words, were the samples scrutinised to ensure that the DNA quality following the extraction was within for example MIQE guidelines levels of quality?
4. On a similar note, the article which the authors refer to in terms of acquiring and preparing the DNA from blood samples mentions that following the acquisition of the blood sample the samples was centrifuged such as to separate the sample into the various cell types. Can the authors confirm which sample population the DNA was extracted from i.e. whole blood, a particular cell types, or other? More details in general are required as to how the samples that were analysed were prepared.
Author Response
We thank the reviewer for the careful reading of the manuscript and constructive remarks. We have taken the comments on board to improve and clarify the manuscript. Please find below a detailed point-by-point response to all comments.
Comments 1: For the control group that is mentioned in the methods, inclusion/exclusion criteria for designating them as the control group should be given in Section 4.1.
Response 1: We revised and added the following sentences in the Methods (lines 357-367):
“For PRS, a control group (n = 515) was selected from individuals within the LGDB resources with available WGS data. These individuals were recruited in LGDB for the Latvian Population Genome Reference project within the framework of the European “1+ Million Genome” initiative. This cohort comprises volunteer Latvian adult citizens, selected to ensure an even distribution of gender, nationality, and place of birth. To maintain genetic variability, participants with first-degree familial relationships were excluded, ensuring no two or more immediate relatives were included. Importantly, we deliberately avoided disease- or phenotype-based exclusion criteria to prevent introducing biases that could result in an artificially reduced or skewed PRS distribution for the selected traits. This approach allows the control group to retain its role as a genetically diverse and representative baseline for PRS evaluation.”
Comments 2: For the blood samples that were utilised as part of the study, more details on the samples and their storage should be given. For example, how long did it take to acquire the samples for the entire cohort, or were these samples that were already taken and the authors granted access to? What was the average age of the samples in terms of storage times? Was the DNA extracted immediately or were the samples stored for a time before the DNA was extracted? Details such as these in terms of the consistency of sampling would be useful.
Comments 3: The authors refer to an article which describes how the DNA was acquired from blood samples that were taken from participants – can the authors confirm that they carried out the same? In other words, were the samples scrutinised to ensure that the DNA quality following the extraction was within for example MIQE guidelines levels of quality?
Response to Comments 2-3: We have supplemented the Methods with a more detailed description of obtaining DNA material from LGDB in lines 369-373:
“The patient's DNA was obtained from LGDB, which specializes in the processing and storage of biological samples (e.g., DNA, plasma, and sera). DNA was extracted from whole blood samples using the phenol-chloroform extraction method, followed by quality control in accordance with LGDB's standard protocols, which comply with OECD and IARC guidelines, as described in detail previously [23].”
We confirm that DNA samples were prepared using the same methodology, described in Rovite et al 2018 work, and in this study we used samples obtained from LGDB.
Comments 4: On a similar note, the article which the authors refer to in terms of acquiring and preparing the DNA from blood samples mentions that following the acquisition of the blood sample the samples was centrifuged such as to separate the sample into the various cell types. Can the authors confirm which sample population the DNA was extracted from i.e. whole blood, a particular cell types, or other? More details in general are required as to how the samples that were analysed were prepared.
Response 4: In line 371, we specified that DNA was extracted from whole blood samples:
“DNA was extracted from whole blood samples using the phenol-chloroform extraction method, followed by quality control in accordance with LGDB's standard protocols, which comply with OECD and IARC guidelines, as described in detail previously [23].”
Round 2
Reviewer 1 Report
Comments and Suggestions for Authors
The manuscript has improved and I thank the authors for addressing my comments.
Author Response
Thank you for your review and for acknowledging the improvements made to the manuscript. We appreciate your feedback, which has helped strengthen our work.
We also have additional clarifications to our manuscript:
We have added units in this sentence in the Results (line 161-163):
“Levels of alanine transaminase were 49 U/L and 33 U/L and levels of aspartate trans-aminase were 44 U/L and 33 U/L (reference <31 U/L for women) with these therapies, respectively.”
Reviewer 2 Report
Comments and Suggestions for Authors
The manuscript was improved. There seemed to be still parts to be revised for improving more the presentation in a scientific manner.
1. In Methods, how was the LDL-C concentration in each patient measured (calculated or directly measured)?
2. Lines 48-55; some references on Latvian people could be cited in the sentences to explain the states.
3. Lines 318-320; some references and concrete data of Latvian people could be added to the sentence to support it.
4. Lines 321-327; some references could be added to the sentence to explain it.
5. Line 353; was the use of the highest/peak LDL-C conducted in the other studies? Please cite similar or referred studies.
6. Line 328-332; why do the authors assume the potential overestimation of LDL-C reported? The evidence may be described in citing some references.
Author Response
Thank you for your review, we greatly appreciate your feedback. Please find our answers below.
Comments 1. In Methods, how was the LDL-C concentration in each patient measured (calculated or directly measured)?
Response 1: Lipid levels were obtained from all previous laboratory reports available at the time of patient inclusion in the Registry. Biochemical measurements (as LDL-C, triglycerides, HDL-C, etc.) were measured by standard state laboratory assays.
We did not gather information on what LDL-C measurement method was used in each of several laboratory reports, but we know that majority of laboratories have been using direct LDL-C measurement method since 2015. Only one large hospital uses either direct or indirect (Friedewald formula) measurement, but outpatients use this laboratory rarely and the Friedwald formula is never used for patients with triglyceride levels >4.5 mmol/L. Other formulas such as Martin-Hopkins or Sampson, have never been used by laboratories in Latvia. Direct LDL-C measurement method has been used in other studies (e.g., by Trinder et al. 2020).
The discrepancies between various indirect methods and direct LDL-C values are mostly seen in patients with higher TG levels and/or low LDL-C levels, and to much lesser extent with higher LDL-C levels [Martin SS, et al. 2013].
We therefore do not expect that in our cohort of patients with very high LDL-C levels a substantial bias due to LDL-C measurement methods.
References:
Trinder, M.; Francis, G.A.; Brunham, L.R. Association of Monogenic vs Polygenic Hypercholesterolemia With Risk of Atherosclerotic Cardiovascular Disease. JAMA Cardiol. 2020, 5, 390–399, doi:10.1001/jamacardio.2019.5954
Martin, S. S., Blaha, M. J., Elshazly, M. B., Brinton, E. A., Toth, P. P., McEvoy, J. W., Joshi, P. H., Kulkarni, K. R., Mize, P. D., Kwiterovich, P. O., DeFilippis, A. P., Blumenthal, R. S., & Jones, S. R. (2013). Friedewald-estimated versus directly measured low-density lipoprotein cholesterol and treatment implications. Journal of the American College of Cardiology, 62(8), 732–739. https://doi.org/10.1016/j.jacc.2013.01.079
We have added the following sentences in the discussion (Lines 335-341):
“We did not gather information on what LDL-C measurement method was used in each of several laboratory reports. However, since 2015, the direct LDL-C measurement method has been used in the majority of Latvian laboratories.
Patients with greater TG levels and/or low LDL-C levels are more likely to have differences between different indirect techniques and direct LDL-C readings, whereas those with higher LDL-C levels are less likely to have these differences [Martin SS, et al. 2013]. Therefore, we do not anticipate significant bias from LDL-C measuring techniques in our group of patients with high LDL-C levels.”
We have added the following sentence in the methods (Lines 362-363):
“Biochemical measurements (as LDL-C, triglycerides, and HDL-C) were measured by standard state laboratory assays.”
Comments 2: Lines 48-55; some references on Latvian people could be cited in the sentences to explain the states.
Response 2: We added the following reference twice in this paragraph (line 48-55):
Latkovskis, G.; Saripo, V.; Gilis, D.; Nesterovics, G.; Upena-Roze, A.; Erglis, A. Latvian Registry of Familial Hypercholesterolemia: The First Report of Three-Year Results. Atherosclerosis 2018, 277, 347–354, doi:10.1016/j.atherosclerosis.2018.06.011.
Comments 3: Lines 318-320; some references and concrete data of Latvian people could be added to the sentence to support it.
Response 3: We revised the following sentence in the Discussion and added the following references (lines 318-321):
“While our study was conducted on a Latvian cohort, we acknowledge that both overall [Reščenko et al.] and FH-related genetic features [Beheshti et al, Alieva et al.] may vary by ethnic groups and the patterns should be replicated in other populations.”
- Reščenko R, Brīvība M, Atava I, Rovīte V, Pečulis R, Silamiķelis I, Ansone L, Megnis K, Birzniece L, Leja M, Xu L, Shi X, Zhou Y, Slaitas A, Hou Y, Kloviņš J. Whole-Genome Sequencing of 502 Individuals from Latvia: The First Step towards a Population-Specific Reference of Genetic Variation. Int J Mol Sci. 2023 Oct 19;24(20):15345. doi: 10.3390/ijms242015345. PMID: 37895026; PMCID: PMC10607061.
- Beheshti, S.O.; Madsen, C.M.; Varbo, A.; Nordestgaard, B.G. Worldwide Prevalence of Familial Hypercholesterolemia: Meta-Analyses of 11 Million Subjects. J. Am. Coll. Cardiol. 2020, 75, 2553-2566, doi:10.1016/j.jacc.2020.03.057.
- Alieva A, et al. European Journal of Internal Medicine 123 (2024) 65–71. https://doi.org/10.1016/j.ejim.2024.01.010
We have no reference that would illustrate how Latvian people genetically differ from other populations in the context of FH. Nevertheless, Reščenko et al. have described previously described genetic diversity of Latvian population.
Beheshti et al. state:
"The exact FH prevalence of a population is dependent on several factors, such as the FH criteria used, ethnicity, age of the population, and others. In this study, we found FH prevalence in the general population in Asia to be 0.19%, compared with prevalence in Europe and North America of 0.32%. This lower prevalence may be due to genetic differences among ethnicities.”
They also argue that:
“Although some Japanese studies use FH diagnostic criteria developed by the Japan Atherosclerosis Society in 2012 (37), many FH diagnostic criteria used in Asia were developed in Western populations, and even though they are often modified, applying these criteria in Asia may not be appropriate, as these populations traditionally have lower cholesterol levels (33)."
Therefore, one may contend that populations with higher LDL-C levels might behave differently when it comes to FH diagnostic criteria or LDL-C cut-off levels.
Alieva et al. showed that “the genetic background of patients clinically diagnosed with FH in two different countries [Russia and Italy] is characterized by high variability.”
Comments 4: Lines 321-327; some references could be added to the sentence to explain it.
Response 4: For the sentences “While the DLCN criteria are a valuable tool for identifying FH, they may have limitations. In younger individuals who may not yet exhibit clinical symptoms like tendon xanthomas or CAD, the FH diagnosis might be underestimated.” (line 322-324) we added the following reference where several authors are experts in the field of FH and FH genetics:
Besseling, et al. European Heart Journal (2017) 38, 565–573; doi:10.1093/eurheartj/ehw135
Here we add some citations from their article:
“Unfortunately, accuracy of current diagnostic criteria is poor, especially in young individuals.”
“Currently, three clinical scoring systems are available (criteria of the Simon Broom registry,15 MEDPED,16 and Dutch Lipid Clinic Network17), and these have been shown to result in mutation detection rates of 61–73%, 53, and 54–70%, respectively, among patients with definitive FH.18 – 20
Unfortunately, these algorithms are crude, especially misclassifying young individuals as being unaffected, while it is eminent that particularly the young benefit most from early detection and initiation of treatment.9 There are two reasons for this misclassification of the young. First, the presence of tendon xanthomas is an important criterion in the Simon Broom and Dutch Lipid Clinic Network criteria, but it is a rare finding in young individuals and therefore may result in misclassifying young HeFH patients as being unaffected. Moreover, fixed, non-age-adjusted cut-off values for LDL-C are used to classify patients. Since LDL-C increases with age,21 this non-age-adjusted approach also contributes to misclassification. As a consequence, in young patients, HeFH is underdiagnosed and undertreated at an even greater extent than in the elderly. Another limitation of the algorithms is that they require information on family history of hypercholesterolaemia and premature CVD, which is often absent,22,23 thus restricting their applicability in clinical practice.”
Xanthomas and arcus cornealis are strong criteria of DLCN, but they usually are not seen in young. The same applies to premature CAD.
For the sentence “Conversely, in a high-risk country like Latvia, premature cardiovascular disease and elevated LDL-C levels could be attributed to lifestyle and other modifiable risk factors, potentially overestimating the likelihood of FH.” (line 324-327) we added the following references:
- Visseren, et al. Eur Heart J. 2021;42(34):3227-3337. https://doi.org/10.1093/eurheartj/ehab484;, the ESC guidelines that classify Latvia as very high-risk country.
- NCD Risk Factor Collaboration Available online: https://ncdrisc.org/country-profile.html (accessed on 13 November 2024). This reference was already used in our manuscript to demonstrate that average cholesterol levels are higher in Latvia than world’s average.
For the sentence “These considerations highlight the need for universal pediatric screening to enhance diagnostic accuracy and identify FH cases early” (line 327-328) we added the following reference:
- McGowan, Mary P et al. Universal paediatric screening for familial hypercholesterolaemia. The Lancet, Volume 403, Issue 10421, 6 – 8. https://doi.org/10.1016/S0140-6736(23)02182-7
Comments 5: Line 353; was the use of the highest/peak LDL-C conducted in the other studies? Please cite similar or referred studies.
Response 5: Yes, this approach has been used in other studies. We added a reference for this sentence “The highest documented LDL-C levels were reported, along with the corresponding TG and HDL-C levels at the time of the peak LDL-C measurement.” (lines 362-365) from the Million Veteran Program where 331 107 multiethnic participants were included:
- Sun YV, et al. Circ Genom Precis Med. 2018 Dec;11(12):e002192. https://doi.org/10.1161/CIRCGEN.118.002192
Here is a citation from the article:
“We examined the individual and collective association between putatively pathogenic FH variants included on the Million Veteran Program biobank array and the maximum LDL-C level over an interval of 15 years (maxLDL). We assessed the collective effect on clinical outcomes by leveraging data from 61.7 million clinical encounters.”
“The MVP participants had a median of 11 measurements (1st quartile of 6 and 3rd quartile of 19) of LDL-C, with 4.4% having only one measurement of LDL-C. When more than one measure was available, we extracted the maximal level of LDL-C to approximate the most likely level of untreated LDL-C”
Additionally, we added the same reference for this sentence (line 329-331), where we mention that we did not conduct standardized lipid measurments, but used highest documented LDL-C levels:
“While we did not conduct standardized lipid measurements at the time of inclusion, accessing historical laboratory reports allowed us to estimate the highest documented lipid levels and identify more cases with pretreatment levels [Sun YV et al].”
Comments 6: Line 328-332; why do the authors assume the potential overestimation of LDL-C reported? The evidence may be described in citing some references.
Response 6: Although we strongly believe that use of maximal or peak LDL-C is the best approach, one should acknowledge the risk of overestimation due to “regression to mean” phenomenon.
Regression to the mean is a statistical phenomenon that occurs when an extreme observation or event is followed by a more typical or average one. In simpler terms, it means that things tend to even out over time. LDL-C levels normally may fluctuate like many other biologic variables.
We added a reference for this sentence “However, the reported peak LDL-C levels may overestimate the average levels due to the regression to the mean phenomenon.” (line 331-333) :
- Regression to the mean: what it is and how to deal with it Adrian G Barnett, Jolieke C van der Pols, Annette J Dobson International Journal of Epidemiology, Volume 34, Issue 1, February 2005, Pages 215–220, https://doi.org/10.1093/ije/dyh299
Additional clarifications:
We have added units in this sentence in the Results (line 161-163):
“Levels of alanine transaminase were 49 U/L and 33 U/L and levels of aspartate trans-aminase were 44 U/L and 33 U/L (reference <31 U/L for women) with these therapies, respectively.”
Round 3
Reviewer 2 Report
Comments and Suggestions for Authors
The paper was much improved.